# Fatigue Crack Growth in a Monocrystal and Its Similarity to Short-Crack Propagation in a Polycrystal of Nickel

**Avihai Petel [1], Ales Jager [2], Dotan Babai [3], Juergen Jopp [4], Arie Bussiba [5], Mordechai Perl [1] and Roni Z. Shneck [3,***

1    Department of Mechanical Engineering, Ben Gurion University of the Negev, Beer Sheva P.O. Box 653, Israel
2    Department of Mechanics, Faculty of Civil Engineering, Czech Technical University in Prague, Thkurova 7, 166 29 Prague, Czech Republic
3    Department of Materials Engineering, Ben Gurion University of the Negev, Beer Sheva P.O. Box 653, Israel
4    Ilse Katz Institute for Nanoscale Science & Technology, Ben Gurion University of the Negev, Beer Sheva P.O. Box 653, Israel
5    Nuclear Research Center Negev, Beer Sheva P.O. Box 9001, Israel
*    Correspondence: roni@bgu.ac.il

**Abstract:** Short fatigue cracks in polycrystalline materials are very important from both practical and basic aspects, yet they are very difficult to observe. Therefore, it is suggested to emulate some properties of short cracks with long fatigue cracks in monocrystals. Indeed, such experiments in pure nickel crystals prove that the three peculiar properties of short fatigue cracks in polycrystalline metals are observed in long cracks in monocrystals, i.e., a lower threshold of the stress intensity factor $(\Delta K_I)_{th}$, a higher crack propagation rate at low $\Delta K_I$ regimes, and the fact that different cracks exhibit different growth behaviors. The fatigue experiments in monocrystals reveal interesting details of the slip activity at the front of fatigue cracks, including a selection rule for the active slip systems above and below the crack, the slip behavior under conditions of steep strain gradients, and the activation of a new slip system.

**Keywords:** short fatigue cracks; fatigue crack propagation; single crystals





## 1. Introduction

Correlating of Fatigue Crack Growth Rate (*FCGR*) with the amplitude of the applied Mode I stress intensity factor ($\Delta K_I$) yields the *FCGR* curve, which exhibits three distinct regions [1,2]: (a) The near threshold region—below the threshold value $(\Delta K_I)_{th}$ no crack growth occurs. Once $\Delta K_I > (\Delta K_I)_{th}$, there is a steep increase in the *FCGR* with an increase in $\Delta K_I$, (b) the Paris' law region—where the *FCGR* curve follows a power law of $\Delta K_I$, and (c) The near critical $\Delta K_I$, where the *FCGR* curve is again a fast-rising function of $\Delta K_I$. All long crack fatigue curves follow this pattern.

Short cracks exhibit three characteristics [3–6]: (a) Short cracks do not have a threshold $(\Delta K_I)_{th}$, below which they do not propagate; (b) the *FCGR* of short cracks is higher than that of long fatigue cracks, (c) different cracks have different crack growth rates within their initial phase of growth. Once the short cracks surpass this initial phase, their *FCGR* converges with the path of the long cracks.

The definition of a short fatigue crack (SFC) remains ambiguous. Ritchie and Lankford [3] listed four possible definitions of a short fatigue crack, of which we find the microstructural definition to be the more significant one. The microstructural definition assumes that the length of the SFC is of the order of a grain size, which implies that an essential part of the "fatigue life" of the short crack is spent within a single grain. Recently, rigorous experimental observations of SFCs have been published (e.g., [7–15]), revealing the tendency of short cracks to propagate along selected crystallographic planes. When the crack encounters a grain boundary, it has to reorient itself and activate different slip systems

in the neighboring grain. This process retards crack growth. The absence of retarding grain boundaries in the path of a SFC at its early stage of growth may explain its higher rate of growth, relative to long cracks. The random encounters of SFCs with grain boundaries due to the randomness of the microstructure may explain the variable and "noisy" character of the curves of SFC growth rate. This variability is also an outcome of the dependence of the *FCGR* in each grain on the crystallographic orientation of the crack [16–20]. The orientation of the crack determines the slip systems that are activated at the crack tip [20–22]. In a recent research on the fatigue behavior of a nickel-based single-crystal superalloy [23], Xiaoyi et al. found that the slip systems and paths that dominate crack extension are altered due to different crystallographic orientations. While the [001] and [314] oriented specimens were mainly dominated by one slip system (planar slip) during fatigue crack extension, the [102] oriented specimens were dominated by two alternating slip systems (cross slip). The transition in slip mechanisms from planar to cross slip is reflected in the fatigue fracture surfaces, which change from flat to rough and in the crack propagation rates, which decreases due to crack deflection and branching in the orientation that allows cross slip.

It is very difficult to follow experimentally a crack in a microscopic single grain within a polycrystalline material due to the irregular shape of the crack. Moreover, its mechanical state is uncontrollable due to the complex mechanical interactions with the surrounding grains. Therefore, we postulate that long fatigue crack growth in a monocrystal, where no grain boundaries exist, can emulate the fatigue growth of a short crack in a polycrystalline material. Under well-defined testing conditions, the fatigue crack growth rate of long cracks in relatively large pure nickel monocrystals was observed and correlated to the fatigue crack growth of short cracks in a polycrystalline material. This experimental model allowed us to explore several questions:

(a) How are the slip systems selected, for each crystallographic orientation of the specimen or the grain, relative to the external load?
(b) To what extent does the crystallographic orientation contribute to the variability of the crack growth rate of SFCs?
(c) How does the crack growth threshold, $(\Delta K_I)_{\text{th}}$, in monocrystals depends on the crystallographic orientation?

## 2. Materials and Methods

Compact tension specimens were cut from commercial polycrystalline pure nickel (200 alloy) and from a monocrystal pure nickel grown in the Institute of Physics, Prague, Czech Republic. The dimensions of the specimens were proportionally reduced to the ASTM E-399 standard, so that the edge length of the specimen was 15 mm and $W$ = 13.5 mm. The initial crack length $a_0 \cong 7$ mm was defined only after the fatigue crack emerged from both surfaces of the specimen (the crack starter was a chevron type made by electro discharge machining. The first specimens were 2.5 mm thick and were found to be extremely ductile. Therefore, the reported experiments were performed on a 5 mm thick specimen in order to suppress plastic deformation. Prior to the test, the specimens were electrochemically polished in a solution containing 70% $H_3PO_4$, 10% $H_2SO_4$, and 20% $H_2O$, under a current density of 4 A/cm$^2$. The polycrystalline specimens were taken parallel to the cross section of an extruded bar, the starting notch was cut in a radial direction, and therefore no texture effect is expected. The average grain size in the cross-section plane was 40 µm. The monocrystal surface was identified by Laue diffraction as the (601) crystallographic plane, deviating 9.5° from the (001) plane. The specimens were cut in two orientations, such that the tensile stress loading will be nearly parallel to the [100] axis or to the [110] axis. Test results for three monocrystals and five polycrystalline specimens are reported herein. The specimens were tested in a Dynamic Fatigue$^{\text{TM}}$ machine. The test frequency was 10 Hz, and the load ratio was $R$ = 0.1. Crack gauges of type KRAK GAGE, made by Rumul Russenberger Prufmaschinen AG company (Neuhausen am Rheinfall, Schweitzer land), were glued on each specimen to monitor the *FCGR* during the test. A fatigue crack was

identified after about $1 \times 10^6$ cycles, and once the crack reached a length of about 8 mm, the test was stopped. Due to the limited number of specimens, only a "near threshold value" was determined using an increasing mode of $\Delta K$. The polished specimen's surface was inspected by OLS50-SU confocal microscope (Olympus, Tokyo, Japan), a scanning electron microscope (SEM) Jeol 5400 (Jeol, Tokyo, Japan), and an atomic force microscope (AFM) MFP-3D-Bio of Asylum Research (Asylum, Santa Barbara, USA).

In order to predict the active slip systems at the crack tip, the stress field at the tip of a sharp crack in a plate under uniaxial tension (known as the Irwin solution [24]) was projected on the 12 slip systems of an FCC crystal (Figure 1) using the stress transformation law:

$$\tau_{RSS} = \sigma'_{kl} = a_{ki}a_{lj}\sigma_{ij} \tag{1}$$

where $a$ is the rotation matrix from the x,y,z system to the coordinate system in which x' is the normal to a slip plane and y' is a Burgers vector on that plane. The resolved shear stresses on the different slip systems were sorted in decreasing order of magnitude (Figure 2). The analysis predicts that different slip systems should operate at different distances and orientations ahead of the crack tip.

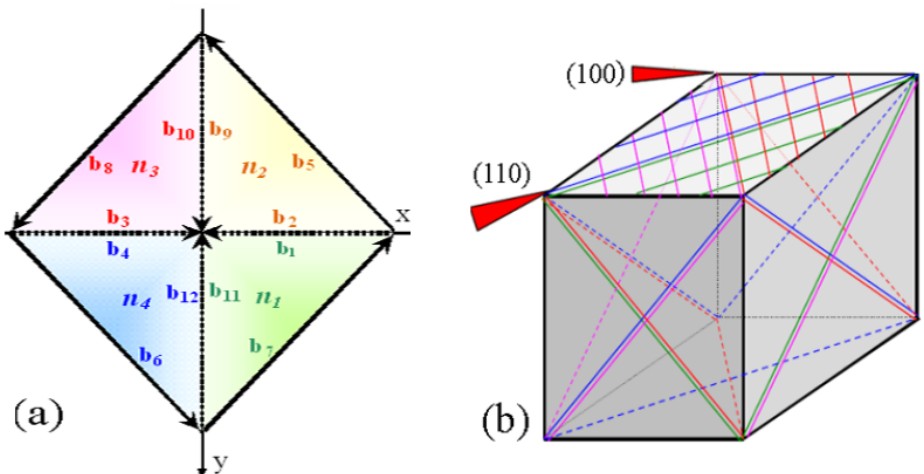

**Figure 1.** The 12 slip systems in an FCC crystal in (**a**) a top-view along the [001] axis, where $\mathbf{n}_i$ are the slip planes and $\mathbf{b}_i$ are the Burgers vectors on each plane. (**b**) The expected intersection lines of the four {111} slip planes with the (001) free surface of the tested crystals are expected to form only two perpendicular traces. The orientation of the two types of cracks is indicated by red triangles.

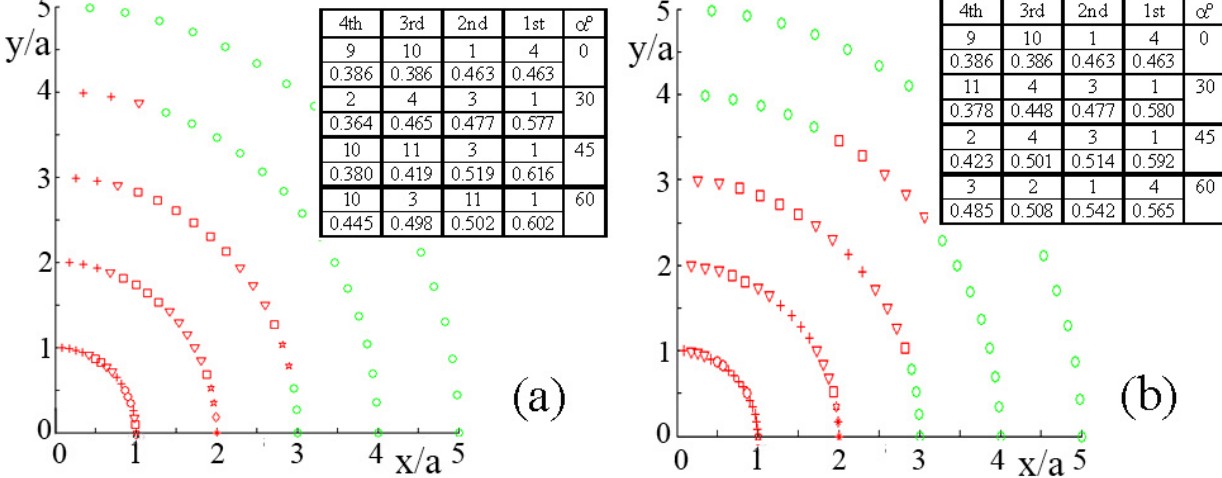

**Figure 2.** The predicted number of active slip systems in front of a sharp crack tip located at the coordinates' origin. The crack with length a is located: (**a**) On the plane $(30, 1, \bar{5})$ (near (100) and tensile

stress along $[\bar{6}, 185, 1]$. (**b**) On the plane $(6, \bar{5}, 1)$ (near $(1\bar{1}0)$ and tensile stress along $[\overline{30}, \overline{37}, 5]$. The symbol "o" indicates single slip, + indicates expected two operative slip systems, a triangle indicates three systems, a square indicates four systems, etc. The tables at the right are the ordered resolved shear stresses on the four most highly stressed slip systems at a constant radial distance *a* from the crack tip, calculated by Equation (1). The resolved stresses are written as multiples of $\sigma^{\infty}$ that is the far field tensile stress applied normal to the crack plane. An active slip system is defined as having a resolved shear stress greater than $0.4\sigma^{\infty}$.

## 3. Results

The *FCGR* in polycrystalline nickel specimens is shown in Figure 3. The *FCGR* of the *slowest* growing crack (*E*) is plotted in Figure 4 along with the *FCGR* in the tested monocrystals. All the monocrystal specimens exhibit a lower $(\Delta K_I)_{\text{th}}$ relative to the polycrystalline specimens. This observed lower threshold for crack propagation may be attributed to the reduced resistance to crack growth in the absence of grain boundaries. Another reason may be that the *FCGR* in monocrystals is intrinsically higher. If this is the real behavior, then the fatigue crack growth curves are expected to be shifted upward and their thresholds are concurrently shifted to the left, i.e., to lower values of $\Delta K_I$ [21]. This observation may hint at the expected similarity between long cracks in a monocrystal and short cracks in polycrystals. Furthermore, a difference in *FCGR* is observed for different crystallographic orientations of the crack plane, despite the fact that the *FCGR* was not the same in repeated tests of specimens with the same crystallographic orientation ((100) crack plane). Notably, the Paris law region of the *FCGR* curves for the polycrystalline specimens is similar in nature to that for the single-crystal specimens. Our fatigue machine enables working in load control mode only when the crack length is relatively short. When the crack length becomes longer, the capability of the machine to keep the selected load lags behind the increasing displacements. The momentary stress intensity factor was calculated using the instantaneous load. As the crack length increases, the applied $\Delta\sigma$ decreases, resulting in a moderate increase of d$a$/d$N$. This moderate change in d$a$/d$N$ was therefore, we believe that in pure constant load conditions the obtained trends will be similar.

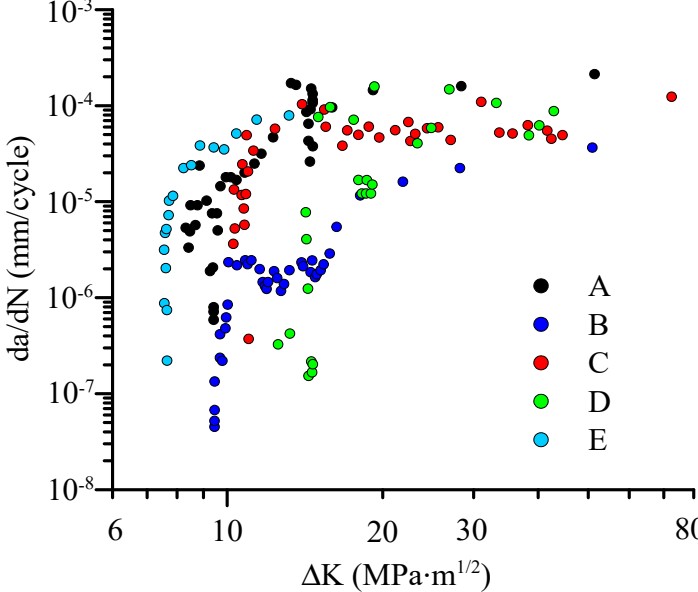

**Figure 3.** FCRG in five *polycrystalline* (PC) nickel specimens (A to E).

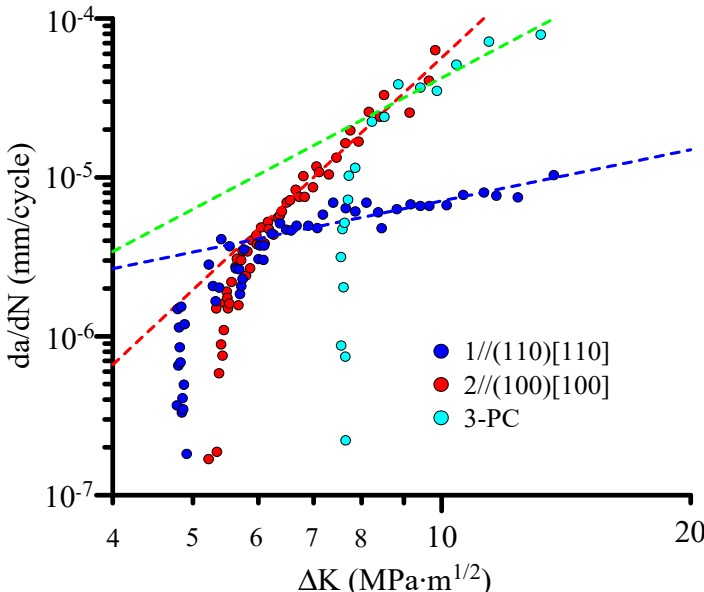

**Figure 4.** FCRG in monocrystals with the fastest-growing crack among the polycrystalline nickel specimens marked by black markers and lines. One crystal with a crack parallel to the (110) plane is designated 1, one crystal with a crack parallel to the (100) plane is designated 2, and the third one is the PC.

Two crystallographic orientations were tested in the present work. In the first two specimens, the tensile axis was [110] and the average crack plane was (110). In the third crystal, the tensile axis was [100], and the average crack plane was (100). The intersections (traces) of two slip planes with the free surface (that is, nearly the (001) plane) took place along the [110] line, and the intersection of the other two slip planes occurred along the $[1\bar{1}0]$ line (Figure 1b). Figures 5–9 show the actually observed slip lines on the free surfaces of the three specimens. As expected, the SEM images in Figures 5 and 6 show that the cracks that propagate along the (110) plane are parallel to the traces of two slip planes and perpendicular to the traces of the other two planes (e.g., Figures 5a,b and 6a,c), while the crack that propagates along the (100) plane bisects the angle between the traces of the slip planes (Figure 7a,e). The crack path tends to follow the slip traces (e.g., Figures 5d,e, 6b,c and 7c), resulting in broken lines (e.g., Figure 5b,c). Since only two perpendicular traces of the known slip planes can be observed on the free surface of the monocrystal cut parallel to a (001) crystallographic plane, the ability to uniquely determine the active slip systems is limited. Nevertheless, it is observed that most of the slip activity on the upper side of the cracks takes place on one slip plane (or two planes with identical traces), while most of the slip on the lower side of the cracks takes place on a different plane, or two planes with identical traces (Figures 5b,d, 6b and 7a,b). This system selection probably reduces the interaction between intersecting slip planes, which could have increased the strain hardening. The selection of a single slip plane on each side of the crack is occasionally violated by a secondary slip (e.g., Figures 5b,d, 6b and 7b,c,e) and in certain patches, traces of nets of two slip traces equivalent in intensity are observed. The two types of traces are perpendicular to each other as expected for the traces of {111} planes on the (001) surface (e.g., Figure 7d,f). This indicates the validity of the analysis presented in Figure 2, which predicted multi-slip conditions at the crack tip. Rarely a third trace is identified, intersecting the {111} traces by 25° (Figures 7c and 9b,d). This third trace is not a {111} plane, which are usually the only operative slip planes in FCC crystals!

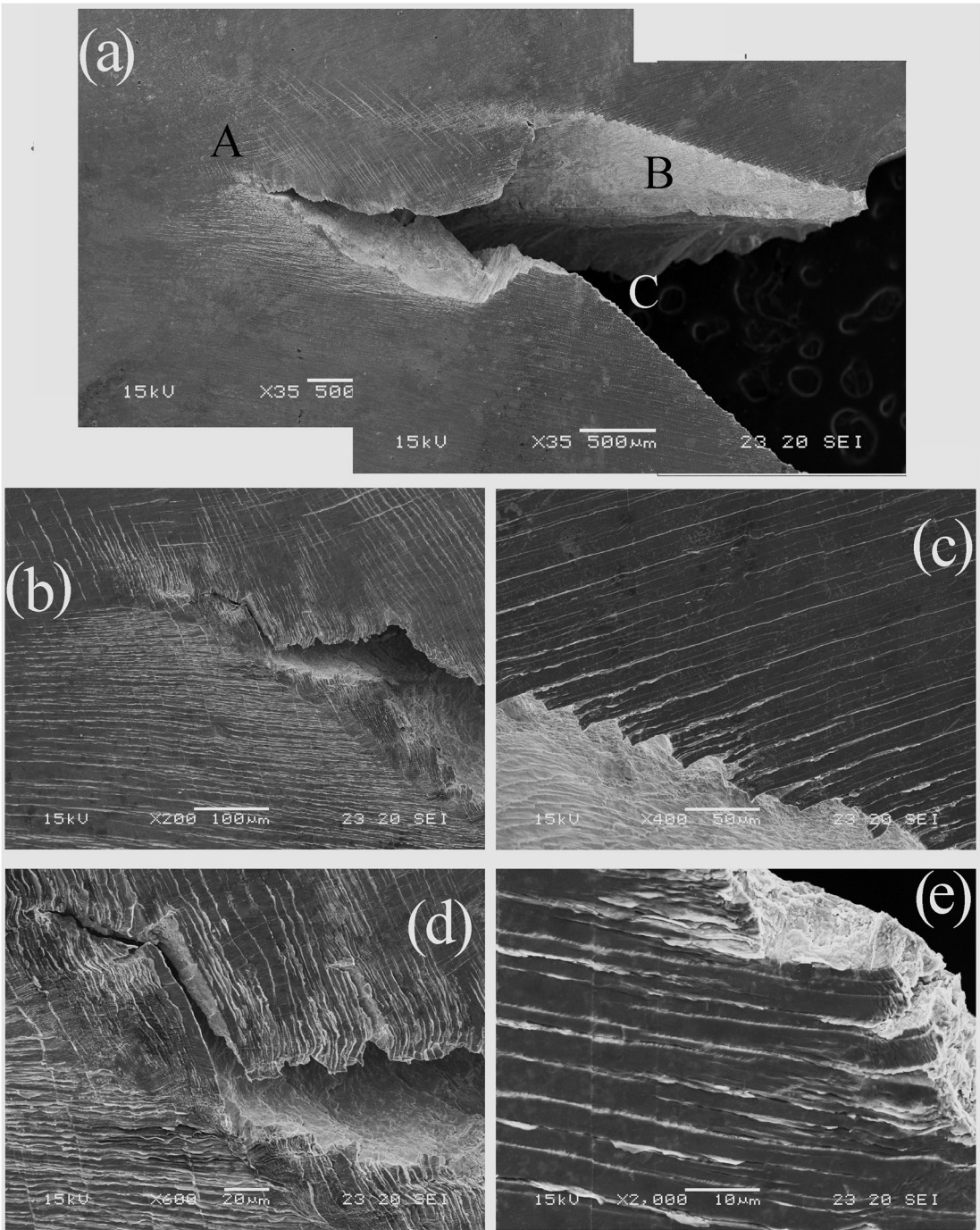

**Figure 5.** SEM images of the free surface of the monocrystal 1 with a crack nearly parallel to the (110) plane. (**a**) general view of the crack, (**b**) the crack tip A, (**c**) area B, (**d**) the tip of the crack in (**b**), and (**e**) area C.

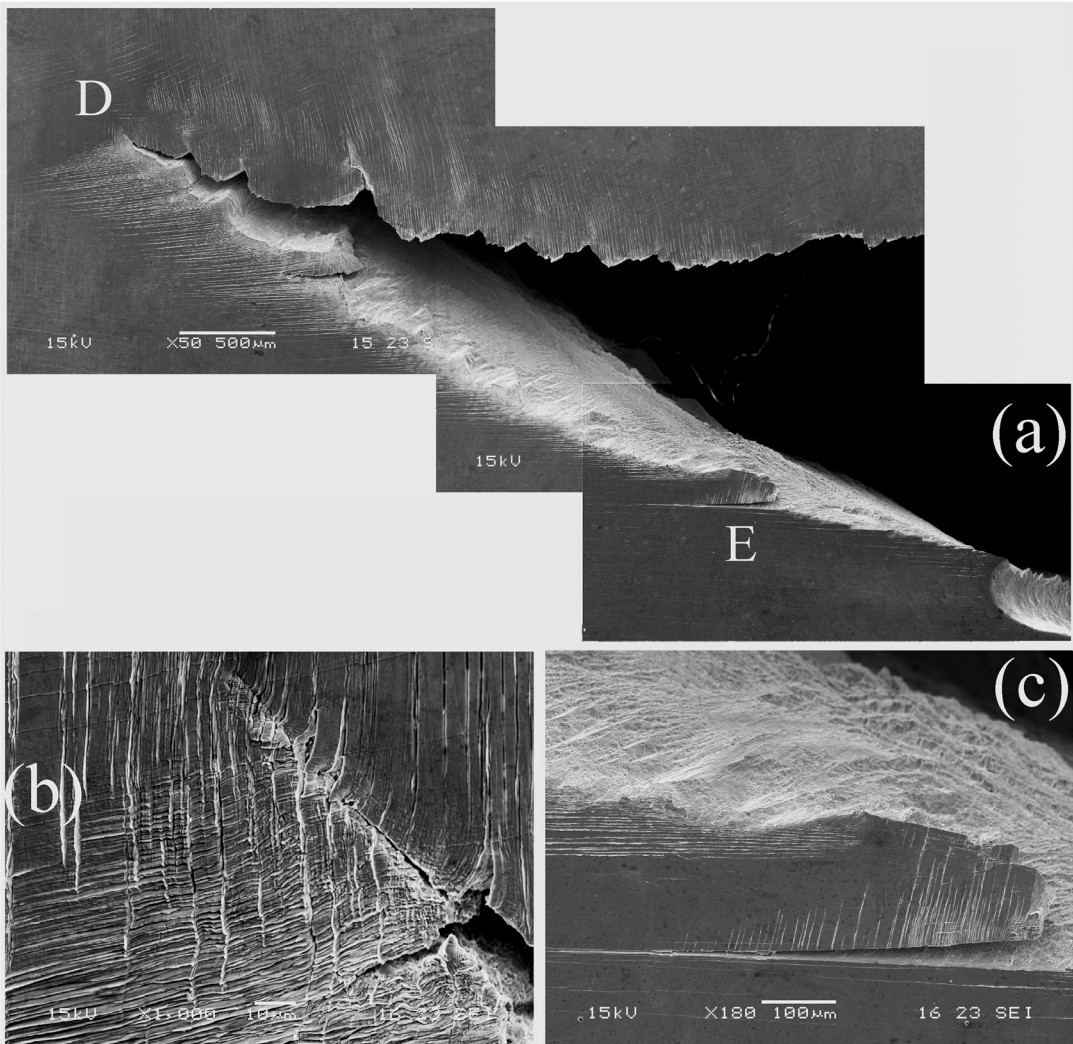

**Figure 6.** SEM images of the free surface of specimen 2 with a crack nearly parallel to the (110) plane.
(**a**) general view, (**b**) the tip of the crack, D, and (**c**) area E.

The slip steps generate extrusions and intrusions, which have attracted much interest in the context of crack initiation but are found to be very common also during crack propagation. Extrusions and intrusions are observed both by SEM and by AFM (Figures 5d,e, 6b, 7g, 8b,e and 9a,b). Slip traces are observed all over the specimens, but their intensity increases in the crack vicinity and as the crack length increases (Figures 5a,b, 6a and 7a,b,e,f), as can be expected from the solution of the stress distribution in front of a crack [23]. In the vicinity of the crack tips, extremely fine and dense slip lines are observed in particular towards the end of the tests (Figures 5b,d, 6b, 7d,f, 8a,d and 9a). The extent of intensive slip is approximately 600 μm around the crack tips. The intensity of slip fades with increasing distance from the crack due to two concurrent changes: the height of the individual slip steps gradually decreases until traces disappear, and the density of slip lines also decreases until only a few high steps remain. This decay of the slip is clearly illustrated in Figures 8a,c,e and 9c. From the observations in Figure 8a,c,e, it can be concluded that the intersection of the slip line, which exhibits weak activity, with more intense lines is the cause for the cessation of the slip on the first slip line, which exhibits weak activity.

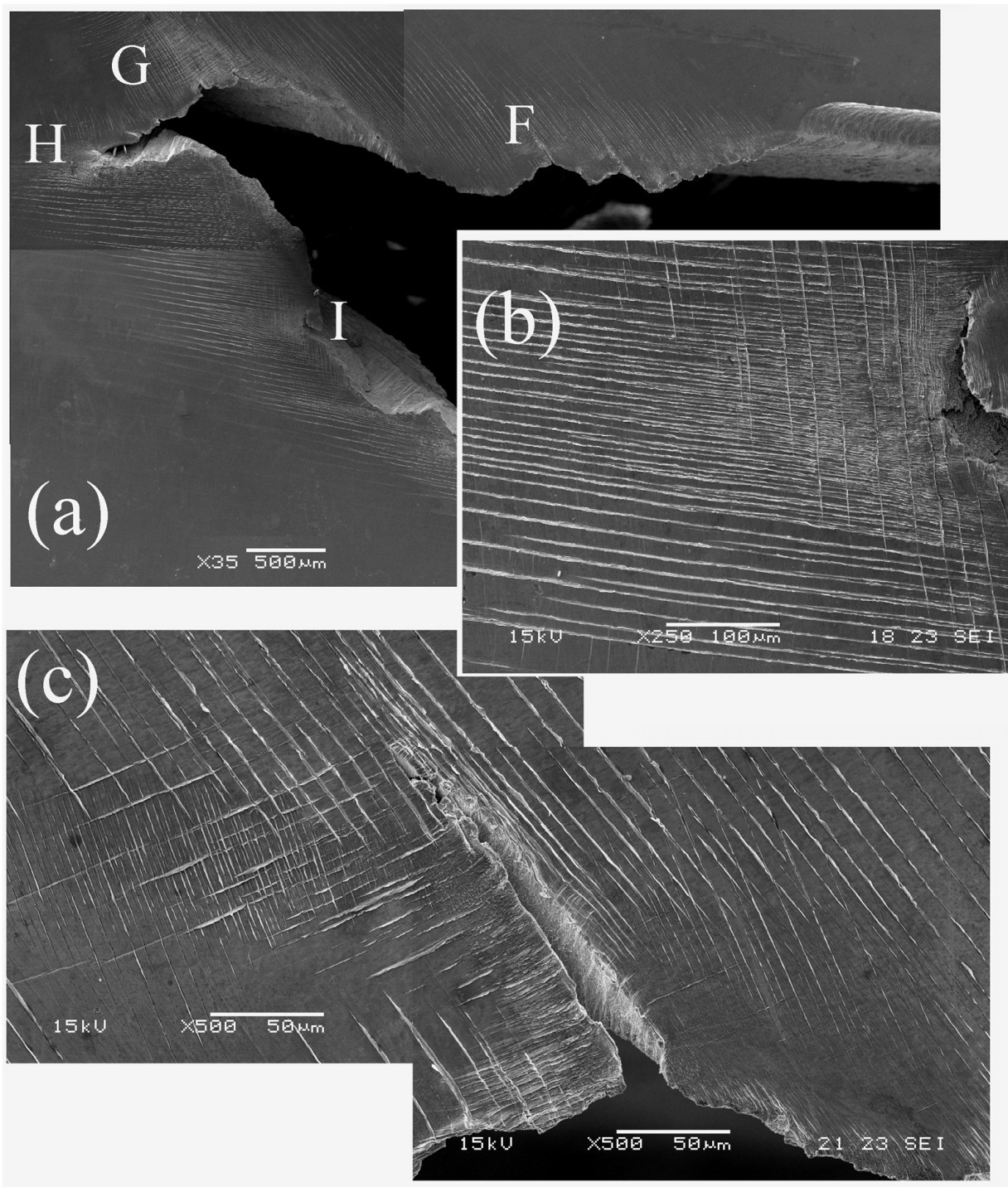

**Figure 7.** *Cont.*

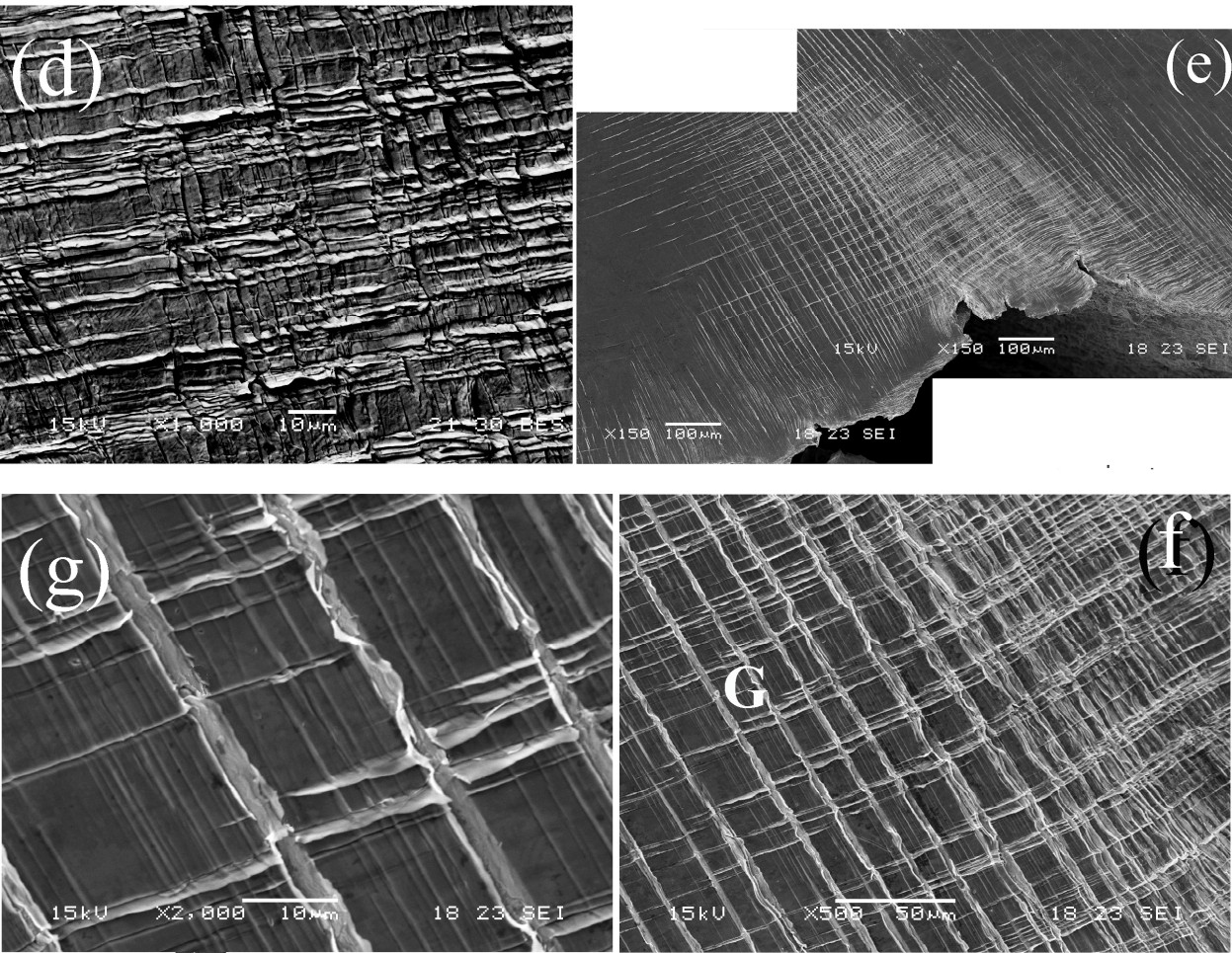

**Figure 7.** SEM images of the free surface of specimen 3 with a crack nearly parallel to the (100) plane. (**a**) general view, (**b**) area I, (**c**) area F, (**d**) area H, (**e**) area G, (**f**) close up in (**e**), and (**g**) area G in (**f**).

The fracture surfaces were observed by optical confocal microscopy and by SEM. Figure 10a shows a transition in the fracture mode of the [110] specimen from planar propagation to a sharp ridge "zig-zag" mode (Figure 10b). In the planar area, lamellae-like brittle fatigue striations were revealed (Figure 10d). The fracture surface of the (100) specimen is planar with well-developed ductile striations (Figure 11).

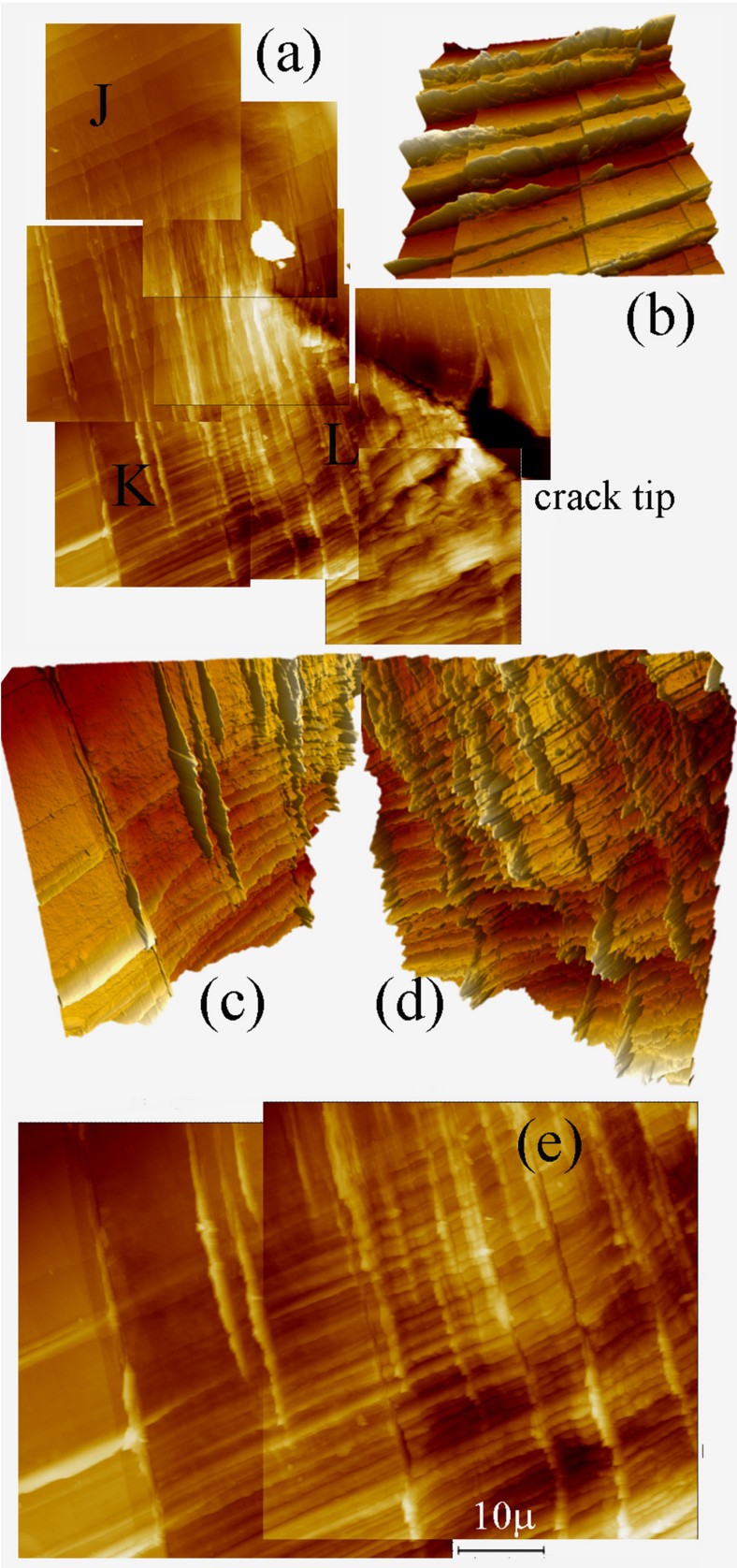

**Figure 8.** AFM images of the free surface of specimen 2. (**a**) General view of the crack tip; (**b**) typical slip steps and extrusions in area J; (**d**) Intensive slip in area L; (**c**,**e**) two views of area K illustrate the fading of the slip lines with increasing distance from the crack.

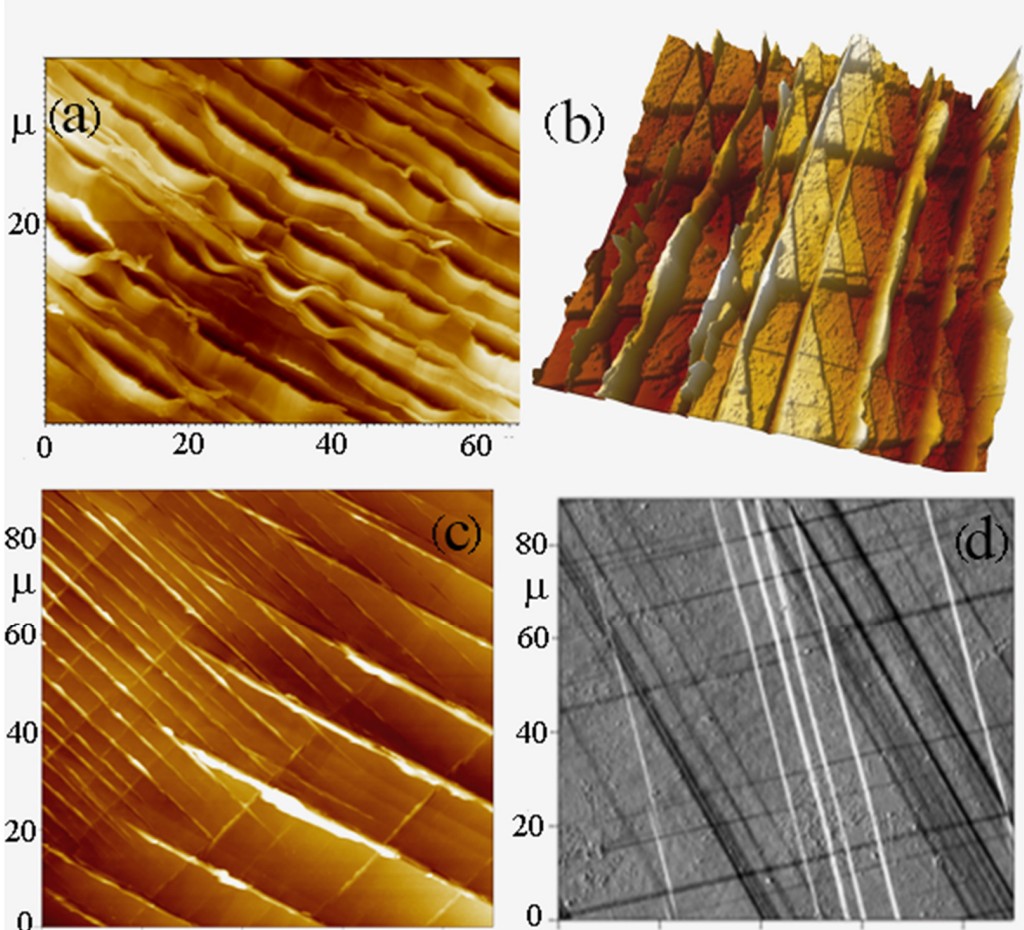

**Figure 9.** AFM images of the free surfaces of specimens 1 and 3. (**a**) intensive slip traces near the crack tip in specimen 1, (**b**,**d**) 3D and 2D views of the activity of three slip systems in specimen 3 area I, and (**c**) fading of the third slip system in area F (Figure 7).

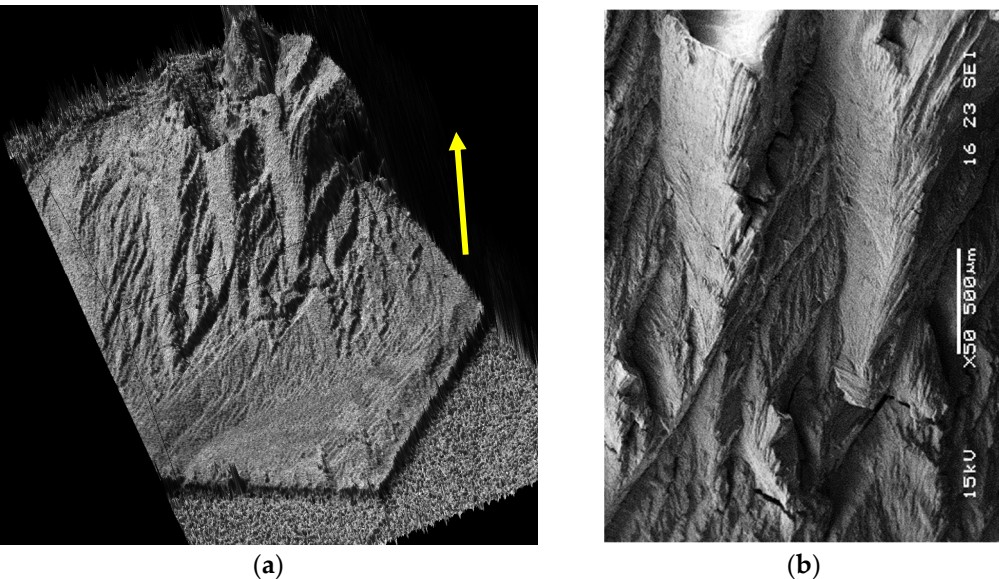

(**a**)  (**b**)

**Figure 10.** *Cont.*

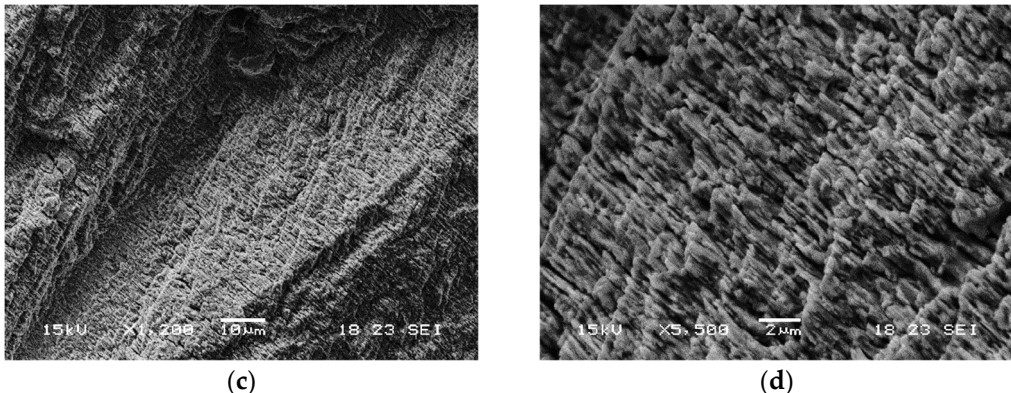

(c) (d)

**Figure 10.** Fatigue fracture modes transition for the [110] crystallographic orientation: (**a**) macroscopic view in a confocal microscope; (**b**) typical ridgs at the second half of the fatigue crack, observed by SEM; (**c**) a facet at the crack initiation zone; (**d**) fatigue crack propagation in cleavage mode. The arrow indicates the fatigue crack propagation direction.

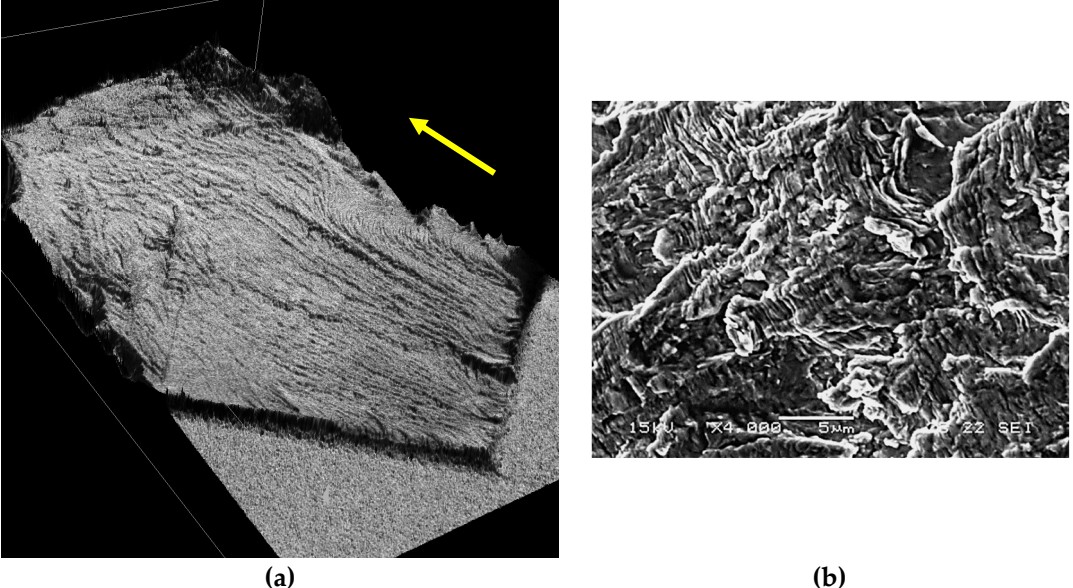

(**a**) (**b**)

**Figure 11.** Fatigue fracture modes in the [100] crystallographic orientation: (**a**) macroscopic view observed by confocal microscope; (**b**) well developed fatigue striations, observed by SEM. The yellow arrow indicates the direction of fatigue crack propagation.

## 4. Discussion

The hypothesis that fatigue growth of a short cracks in a polycrystalline material can be emulated by studying the growth of long fatigue cracks in a monocrystal has been tested in the present work in pure nickel crystals. It can be expected that as the grain size of the polycrystalline material increases, the similarity will increase. The three mechanical characteristics of short fatigue crack growth in polycrystalline metals were observed in the growth of long cracks in single crystals. Namely,

a. The long fatigue cracks in the nickel monocrystal exhibit lower threshold values $(\Delta K_I)_{th}$ than in polycrystalline nickel, similarly to short fatigue cracks in a polycrystalline material.

b. The fatigue crack growth rate (*FCGR*) of a long crack in monocrystalline nickel at low values of $\Delta K_I$ was found to be higher than in the corresponding section of the *FCGR* curve in polycrystalline nickel. This replicates the well-known behavior of short fatigue cracks in polycrystalline metals.

*FCGR* curves of short cracks are known to be erratic [3,4,7,10–12,25], due to the randomness of the material microstructure and the short cracks' random encounters with grain boundaries. Likewise, long cracks propagating through the monocrystal at different crystallographic orientations exhibited diverse *FCGR* curves and fracture morphology.

In view of these findings, we suggest that the fatigue crack growth of long cracks in monocrystals can be used as a convenient model system for the FCG of short cracks in polycrystalline materials.

The crystallographic orientation can affect the crack growth rate via crack deflection, branching, and crack closure induced by the roughness of the crack surface. Following the fractographic observations in Figures 10 and 11, one can draw the following conclusions:

a.  The crack path in the {110} plane (Figure 10) is characterized by roughness and deflections. The P-COD loop, not presented, indicates the existence of significant crack closure resulting from the high level of fracture roughness. They reduce the crack driving force and hence the effective stress intensity factor amplitude, $\Delta K_{I\text{eff}}$, giving rise to a slower crack growth rate and a relatively flat Paris slope ($n = 1.1$) (Figure 4).

b.  The crack path in the {100} plane is smooth (Figure 11). Crack roughness and deflections are smaller, and $\Delta K_{I\text{eff}}$ is high. This gives rise to a fast crack growth rate and steep slope off the Paris slope ($n = 4.9$) (Figure 4).

c.  The present results are in accordance with other observations, e.g., Chan et al. [16] who performed fatigue tests on a monocrystal of Mar-M200 with different crystallographic orientations. According to Xiaoyi et al. [23], orientations close to (110) activate two competing slip systems, and the path of the growing fatigue crack deflects and branches. These extrinsic phenomena reduce the fatigue crack growth rate and increase the total fatigue life. On the other hand, orientations close to (100) activate mainly planar slip. The *FCGR* is higher since no extrinsic sources are being activated, which could have delayed crack extension.

In FCC crystals pure tension along the [100] activates eight slip systems with equal resolved shear stresses. The orientation [110] activates only four slip systems in FCC crystals at pure tension. The resolved shear stress in front of the Irwin crack strongly varies with the inclination relative to the crack line. Our analysis in Figure 2 shows only a small difference between the two orientations, with slightly higher resolved stresses and a higher number of activated slip systems in the (100) orientation. The simple analysis does not predict the above-mentioned single slip situation in the (100) orientation and double slip in the (110) orientation. Yet, the macroscopic shape of the (100) fracture surface is smooth, clearly different from the sharp ridge topography developed at an intermediate stage of the crack propagation in the (110) orientation and associated with a lower *FCGR* and a smaller $\Delta K_{I\text{th}}$. More work is needed to identify the actual slip directions and explain the difference in the *FCGR*s.

Several interesting metallurgical observations regarding the slip activity at the front of fatigue cracks in a monocrystal were made:

a.  Different slip planes are dominant above the crack plane and below it. This selection rule probably reduces the strain hardening during crack growth.

b.  A new slip system (none {111}) is occasionally activated whenever intensive slip is required.

c.  In regions of high stress intensity factors, namely close to the crack-tip, and as crack length increases, the slip intensity increases by activation of dense slip. The intense shear gradually fades by reducing both the height and the density of the active slip planes.

Further experimental work is necessary to propagate long fatigue cracks in other metal monocrystals to further strengthen and generalize the present findings.

## 5. Conclusions

Mechanical and metallurgical aspects of fatigue crack growth in nickel monocrystals were explored:

Mechanical behavior: the three characteristic properties of short fatigue cracks in polycrystalline metals are properly emulated by long fatigue cracks in monocrystals, namely:

(a)    Different cracks exhibit different *FCGR* behavior.
(b)    The *FCGR* at low $\Delta K$ of a long crack in a monocrystal is higher than the corresponding part of the *FCGR* curve of a polycrystal.
(c)    Long fatigue cracks in a monocrystal exhibit lower threshold $\Delta K_{th}$. values.

Therefore, we suggest that *FCGR* of long cracks in monocrystals may be used as a model system for *FCGR* of short cracks in polycrystalline materials.

The difference between the smaller *FCPR* of the (110) orientation relative to the (100) one is explained in terms of deflection, branching, on the microscopic scale, and crack closure, on the macroscopic scale, being more dominant in cracks growing parallel to the (110) plane. This difference may be an outcome of double slip vs. single slip orientations, according to ref [23].

Slip behavior: new interesting details of the slip activity at the front of fatigue cracks in a monocrystal were observed, namely:

(a)    Different slip planes are dominant above and below the crack plane. This selection rule apparently reduces the strain hardening during crack growth.
(b)    A new slip system (none {111}) is occasionally activated whenever intensive slip is required.
(c)    In regions of high stress intensity factors, namely close to the crack tip and with increasing crack length, the slip intensity increases by the activation of dense slip planes. The intense shear gradually fades by reducing both the height and the density of the slip steps.

**Author Contributions:** A.P.: methodology and investigation; D.B.: software, data curation, and validation; A.J.: growth of the Ni single crystals; J.J. performed the AFM observations; A.B. and M.P.: supervision and writing; and R.Z.S.: conceptualization, supervision, and writing. All authors have read and agreed to the published version of the manuscript.

**Funding:** This research received no external funding.

**Institutional Review Board Statement:** Not applicable.

**Informed Consent Statement:** Not applicable.

**Data Availability Statement:** Additional data is available in the MSc thesis of Avihai Petel at the Ben Gurion University of the Negev, Beer Sheva, Israel.

**Acknowledgments:** We acknowledge the technical assistance of Igal Alon in conducting the instrumented fatigue experiments.

**Conflicts of Interest:** The authors declare no conflict of interest.

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
