# Peer review of "Fatigue Crack Growth in a Monocrystal and Its Similarity to Short-Crack Propagation in a Polycrystal of Nickel"

_metals, doi:10.3390/met13040790_

Round 1

Reviewer 1 Report

The submitted paper performed several experiments with growing fatigue cracks in nickel single and polycrystals with the aim to clarify the mechanism of short crack growth in polycrystals. They drew some conclusions on the emulation of growth of fatigue cracks in polycrystals using their results and also some findings concerning the slip activity at the front of fatigue cracks in single crystals.

Critical comments:

1.     Authors should consider the change of the title of the contribution to reflect actual content.

2.     The dimensions of the specimens are not given precisely enough.

3.     p. 2, l. 77 Ductility of the material is not function of the thickness of the specimen. It is thus surprising that 5 mm thick specimen was OK while 2.5 mm not.

4.     How was the shape of the crack front in single – and polycrystals. How the crack length a was defined in both cases?

5.     Fig. 1 needs more commentary.

6.     Fig. 2: Smidt factors higher than 0.5 need commentary and explanation. is not defined.

7.     Fig. 3: What was the crack length of the "short cracks"? How was the crack length evaluated? How big was the cyclic plastic zone?

8.     p. 5: Descriptions of the crack path in single crystal given here hardly emulate the real crack path inside the individual grain in a polycrystal. Has authors some proof of this?

9.     p. 9: Slip activity on the surface and extrusions and intrusions is only a sign of cyclic plastic zone and relation to the crack growth mechanisms is not given. Figs. 7 to 9 are not relevant to the subject.

10.  p. 11, l. 213-216 "FCGR curves of short cracks are known to be erratic, due to randomness of the material microstructure and the short cracks’ random encounters with grain boundaries." This is not true, see e.g. J. Polák et al. Procedia Engineering 2(1) (2010) 883-892. There is some scatter but systematic dependence on the crack length and plastic strain amplitude is evident.

11.  The conclusions of the paper are either trivial or do not reflect the results achieved. They should be concentrated on the subject of the mechanism and kinetics of the crack growth - implications of the crack growth in single crystals on that of polycrystals.

Reviewer 2 Report

The authors emulated the fatigue growth of short crack in polycrystalline materials to long cracks in mono crystalline one. Fractography observations were reported to explain the mechanism of fatigue crack growth. Some questions need to be answered.

1-     The necessity of this emulation of fatigue crack in a polycrystalline with a long crack in a mono crystalline is not well addressed.

2-     The tables in figure 2 are not clear. More explanation is needed. (it is clear from the text that it is the Schmidt factor for slip systems, but it could be better explained in the figure).

3-     In the polycrystalline, it is important to know whether a texture existed or no, as it can affect the crack growth behaviour. Did the authors do that?

4-     In a poly crystalline, we cannot say which crystallographic direction a crack is present due to the randomness of the grains. How the authors did this?

5-     The authors did not provide enough evidence if a new slip system other than 111 was activated.

Reviewer 3 Report

Comments Manuscript:  Emulating Growth of “Short Fatigue Cracks” in a Polycrystalline Material Using “Long Crack” in Monocrystalline Specimens

Dear authors,

Thank you for your interesting and sound work in your manuscript.

Comments about the experimental methods.

1)   It would be interesting/necessary to mention the grain size of the polycrystalline material that was tested.

2)   Some relevant details about the fatigue tests are missing, i.e., test frequency and load ratio.

3)   Please, specify the crack gages used (brand) and more details about the expected accuracy/resolution of the crack length measurement.

4)   Please specify the brand and type of fatigue machine used.

5)   Please, describe if and how a notch was created, and the geometry of the notch.

6)   Please provide figures 5, 6 and 7 with an arrow indicating the loading direction.

Please address the following comments:

1)   Fatigue crack growth in a large a single crystal is measured, aimed to eliminate the influence of grain boundaries, i.e., to gain a more fundamental understanding of fatigue crack growth. Indeed this has been achieved. However, in polycrystalline materials small crack and long cracks (with the same role of grain boundaries) are also known to show different crack growth rates, which is often ascribed to crack closure to develop for long cracks especially.
In the manuscript the role of crack closure is mentioned but not analyses in detail. Therefore, the effect of grain boundaries cannot be isolated from the results. Can you elaborate on this matter please?

2)   Normally, determining Delta(K) threshold values require a special procedure growing a crack at a certain Kmax level, and incrementally increasing Kmin , with resulting decrease in Delta(K), until da/dN drops below a critical value. Was this procedure applied?

3)   Figure 3 shows the “Paris curves” of 5 replicate tests of the polycrystalline nickel, with the following remarks:
a) the results show pronounced scatter. Please motivate why these results are still considered accurate enough to draw conclusions.
b) There seems to be a very limited increase in da/dN
 with increasing Delta(K) Please elaborate on this.

Round 2

Reviewer 1 Report

I have read the revised version of the manuscript. In most aspects, it is improved enough to guarantee my recommendation for publication. 

Author Response

Thank you for accepting our reply.

Reviewer 3 Report

Dear authors, 

Thank you for addressing my comments in the paper, it is really appreciated. Nevertheless, I still not fully understand the explanation about figure 3. I can see that load levels could not be fully applied for longer cracks due to the lower stiffness of the specimen, and the probably limited capability of the fatigue machine. However, this should not affect the Paris curve if actual load range levels are used calculate the stress intensity range. I just wanted to mention this. 
